# Non-invasive adapted N-95 mask sampling captures variation in viral particles expelled by COVID-19 patients: Implications in understanding SARS-CoV2 transmission

Kalpana Sriraman[1☯], Ambreen Shaikh[1☯], Swapneil Parikh[2], Shreevatsa Udupa[2], Nirjhar Chatterjee[2], Jayanthi Shastri[2,3‡], Nerges Mistry[1‡]*

1 The Foundation for Medical Research, Mumbai, Maharashtra, India, 2 Molecular Laboratory, Viral Research, and Diagnostic Laboratory, Kasturba Hospital for Infectious Diseases, Sane Guruji Marg, Mumbai, Maharashtra, India, 3 Department of Microbiology, BYL Nair Charitable Hospital, Mumbai, Maharashtra, India

☯ These authors contributed equally to this work.
‡ These authors are joint senior authors on this work
* fmr@fmrindia.org

**Data Availability Statement:** All relevant data are within the manuscript and its Supporting Information files.

## Abstract

Infectious respiratory particles expelled by SARS-CoV-2 positive patients are attributed to be the key driver of COVID-19 transmission. Understanding how and by whom the virus is transmitted can help implement better disease control strategies. Here we have described the use of a noninvasive mask sampling method to detect and quantify SARS-CoV-2 RNA in respiratory particles expelled by COVID-19 patients and discussed its relationship to transmission risk. Respiratory particles of 31 symptomatic SARS-CoV-2 positive patients and 31 asymptomatic healthy volunteers were captured on N-95 masks layered with a gelatin membrane in a 30-minute process that involved talking/reading, coughing, and tidal breathing. SARS-CoV-2 viral RNA was detected and quantified using rRT-PCR in the mask and in concomitantly collected nasopharyngeal swab (NPS) samples. The data were analyzed with respect to patient demographics and clinical presentation. Thirteen of 31(41.9%) patients showed SARS-COV-2 positivity in both the mask and NPS samples, while 16 patients were mask negative but NPS positive. Two patients were both mask and NPS negative. All healthy volunteers except one were mask and NPS negative. The mask positive patients had significantly lower NPS Ct value (26) compared to mask negative patients (30.5) and were more likely to be rapid antigen test positive. The mask positive patients could be further grouped into low emitters (expelling <100 viral copies) and high emitters (expelling >1000 viral copies). The study presents evidence for variation in emission of SARS-CoV-2 virus particles by COVID-19 patients reflecting differences in infectivity and transmission risk among individuals. The results conform to reported secondary infection rates and transmission and also suggest that mask sampling could be explored as an effective tool to assess individual transmission risks, at different time points and during different activities.

**Funding:** This study was supported by grants and donations from Godrej Agrovet Limited-Mumbai, Zoroastrian Charity Funds of Hong Kong, Canton and Macao, and The Vashketu Foundation-Mumbai. The funders of the study had no role in study design, data collection, data analysis, data interpretation, or writing of the manuscript.

**Competing interests:** The authors have declared that no competing interests exist.

## Introduction

One year into the COVID-19 pandemic, there have been over 100 million confirmed cases and over 2 million deaths due to COVID-19 worldwide. SARS-CoV-2 spreads more easily compared to SARS-CoV-1 and MERS-CoV as reflected by a higher R0 and higher household secondary attack rate [1,2]. The dispersion factor for COVID-19 has been estimated to be as low as 0.1 indicating that COVID-19 transmission is over-dispersed, which means a small number of infected individuals drive most of the spread [3]. The transmission is driven by super spreading events that occur due to the interaction of a host, an agent, and environmental factors. Identifying patient characteristics that correlate with super spreading might allow focused and targeted non-pharmaceutical interventions to bust COVID-19 clusters and contain the spread.

There is an emerging consensus that the bulk of transmission occurs when infectious individuals with COVID-19 generate respiratory particles of varying size, which are airborne over varying distances and time, and are inhaled by susceptible individuals, resulting in the transmission of SARS-CoV-2 [4–6]. Collecting nasopharyngeal or oropharyngeal specimens by inserting a swab may not correlate with the potential of the host to generate infectious respiratory particles, nor reflect different host activities that result in different transmission risks; singing and heavy breathing during exercising are thought to result in more infectious particles than speaking softly or quiet breathing [7,8]. Thus there is a need for sampling methods that better reflect the transmission risk of infected individuals particularly during different actions such as breathing, speaking, shouting or singing in different hosts.

Various studies conducted during flu seasons have shown the feasibility of detecting viruses in exhaled breath condensates using commercially available bio-samplers and cough sampling systems [9–11]. Even face mask sampling–a low-cost method–has also proved to be effective for analyzing exhaled/expelled respiratory particles and detecting respiratory pathogens like the influenza virus [12,13]. Our earlier work has demonstrated that respiratory particles captured on a membrane attached to N-95 masks worn by patients of tuberculosis (TB), another air-borne disease, can be used to detect and isolate viable TB bacterial RNA in a noninvasive manner with 96% accuracy [14]. COVID-19, like TB, is predominately transmitted by infectious respiratory particles and hence we hypothesized that this method may be adapted to detect SARS-CoV-2 for applications in diagnosis and understanding risks of transmission from COVID-19 patients. In this study, we demonstrate that our mask sampling method can be used to detect SARS-CoV-2 RNA generated by COVID-19 patients using real-time reverse transcriptase-polymerase chain reaction (rRT-PCR), and the cycle threshold (Ct) value can indicate the potential infectiousness of different patients [15]. This method may have important applications in studying variations in infectiousness between patients and in the same patient during different activities that would help assess the transmission risk.

## Materials and methods

### Patient recruitment and sample collection

The study was undertaken between June and September 2020 after approval of the Institute Research Ethics Committee of The Foundation for Medical Research (FMR) (FMR/IREC/TB/01/2020), Mumbai, and the Institutional Review Board of Kasturba Hospital for Infectious Disease, Mumbai (IRB-09/2020). Thirty-one adult symptomatic patients with mild/moderate COVID-19 admitted to the COVID care ward in Kasturba Hospital were enrolled in the study after taking written informed consent. The SARS-CoV-2 positivity was confirmed either by rapid antigen test or oropharyngeal swab–rRT-PCR test. An equal number of asymptomatic

healthy volunteers with no known contact with COVID-19 patients were enrolled as controls in the study at FMR after taking informed consent. The sample size was calculated using a proportion test for binary outcome with assumptions of 95% confidence interval, 80% power and 10% acceptable difference. Demographic characteristics, clinical presentations, and treatments were recorded for all the study participants.

A mask sample and a nasopharyngeal swab sample (NPS) were collected from each of the patients and healthy volunteers. For patients, the samples were collected within 36 hours of their confirmed diagnosis. For mask sampling, participants wore a modified cup-type N95 mask (Venus Safety and Health Private Limited, Navi Mumbai, India) with an attached commercially available 37mm diameter gelatin membrane (Sartorius, Gottingen, Germany, Supplementary Fig S1 in S1 File) on the inner surface of the mask for 30 minutes. The participants were asked to carry on with the activities whatever they were doing for the first 20 minutes and undertook certain purposeful vocal tasks in the last 10 minutes. The purposeful tasks included following tasks in sequence as directed by the sample collector.

i.  Talk or Read—3 mins

ii.  Cough 20 times- (1 minute)

iii.  Deep breath for 1 minute

iv.  Talk or Read-3 mins

v.  Cough 20 times- (1 minute)

vi.  Deep breath for 1 minute

After completion of mask sampling, the membrane was removed from the mask using sterile disposable forceps and transferred to a collection cup containing 3ml of RNAzol™ (Sigma-Aldrich, MO, USA). The collected sample was then transported to the FMR laboratory at room temperature for further processing. During mask sampling, the sample collector subjectively noted the actual intensity with which, each participant performed the vocal task and recorded the details in the questionnaire format of the case record form (Supplementary information- mask sampling section). The quality of sampling was measured by assigning a sampling score for each activity based on the intensity of the task. The following scoring pattern was used for the 3 tasks- Loud talking/reading = 3, Normal talking/reading = 2, low talking/reading = 1, Deep and forceful continuous coughing = 4, deep and forceful intermittent coughing = 3, light and continuous coughing = 2, light, and intermittent coughing = 1, deep breathing = 2, shallow breathing -1. A retrospective analysis of the human RnaseP gene, an indicator of sample quality was carried out in all mask samples using TaqPath SARS-CoV-2 detection kit V1 (Details in supplementary information)

Following mask sampling, an NPS was collected from the patients. The swab was collected in viral transport media (HI Viral transport kit, HiMedia Laboratories, Mumbai, India), and transported to Kasturba laboratory at 4˚C for further processing. For NPS, ICMR approved standard protocols and rRT-PCR were used for RNA extraction and detection of SARS-CoV-2.

## Sample processing and quantitative real-time PCR

Total RNA was isolated from 3ml RNAzol™ containing dissolved gelatin membrane as per the manufacturer's protocol. Internal Control (IC) and carrier RNA were added to the RNAzol sample before isolation. The RNA obtained was purified using QIAamp viral RNA isolation kit (Qiagen, Hilden, Germany).

The rRT-PCR was carried out in CFX 96 real-time thermal cycler (Bio-Rad Laboratories, California, USA) and SARS-CoV-2 genes were detected using RealStar® SARS-CoV-2 RT-PCR Kit (altona Diagnostics, Hamburg, Germany) as per the manufacturer's protocol. The kit detects the E gene for *betacornoviridae* and the S gene specific for SARS-CoV-2. The positive control used was part of the detection kit, while the negative control was RNA isolated from TB patients using mask aerosol sampling, collected before December 2019 (Pre-COVID). As the patient samples were from confirmed COVID-19 patients, the detection of both E and S genes or either E gene or S gene with visible sigmoidal PCR amplification curves were considered positive. All mask samples collected from healthy volunteers were also tested for SARS-CoV-2 using the same protocol. To determine the viral copy numbers from SARS-CoV-2 positive aerosol samples, a standard curve was generated from 10-fold serial dilutions of the SARS-CoV-2 E gene (included in SARS-CoV-2 Positive material IVT kit, Supplementary Fig S2 in S1 File) and analyzed using RealStar® SARS-CoV-2 rRT-PCR assays.

## Statistical analysis

The results were statistically analyzed using Graph Pad Prism software (version 6.01). Percentages were calculated for categorical variables, and statistical significance was assessed using $\chi^2$ and Fisher exact tests. For continuous variables, the median with interquartile range (IQR) was calculated, and statistical significance was assessed using Mann Whitney unpaired t-test, and a p-value of $< 0.05$ was considered significant.

## Results

Of the 31 previously confirmed COVID-19 patients, SARS-CoV-2 viral RNA was detected by rRT-PCR in 29 (93.54%) NPS samples while expelled SARS-CoV-2 virus was detected in mask samples of 13 patients (44.8% of contemporary NPS positive patients and 41.9% of 31 confirmed patients). For two patients the virus was neither detected in NPS nor in mask samples collected at the time of enrollment. Among 31 healthy volunteers, one asymptomatic person was positive by NPS sampling but negative by mask sampling, while all others were negative by both NPS and mask sampling. The mask samples were assayed for two target SARS-CoV-2 genes (E and S). Both these genes were detected in 11 of the 13 patient samples, while 2 samples were only positive for the E gene. The Ct values for the mask positive patient samples had a median value of 36.97 (IQR 32.50–38.01) for the E gene and 35.73 (IQR 31.27–39.15) for the S gene. The Ct of the mask samples in patients was significantly higher (p = 0.0010) than the corresponding NPS samples.

We grouped the patient data into mask positive and mask negative patients and compared patient characteristics, SARS-CoV-2 specific variables, symptoms, and qualities of mask sampling (Table 1). Mask positive patients had significantly lower (p = 0.008) NPS Ct values (median value 26, IQR 21–29.5) than mask negative patients (median value 30.5, IQR 28–32). Mask positivity in patients was associated with higher rapid antigen test positivity in NPS samples at diagnosis (p = 0.025), the likelihood of having contracted the disease from a known contact (61.5% mask positive patients had known contact vs 37.5% in mask negative patients), and likely to have fever as a symptom (100% mask positive patients with 46% having high fever vs mask negative patients with 69% fever and 6% having a high fever). There were no significant differences in other symptoms, characteristics, or treatment. Since the respiratory output is linked to intensities of various vocal and respiratory activities [16], we determined the quality of sampling based on an assigned sampling score as described in the methods. We observed that mask positive patients had a median sampling score of 8 (IQR 5.5–8) while mask negative patients had a score of 6 (IQR 5.2–7). The variation in the sampling score was not significant,

**Table 1. Comparison of nasopharyngeal swab Ct, symptoms, treatment and mask sampling characteristics among mask positive and mask negative patients.**

| Descriptions | NPS Positive (n = 29)* | | | | Healthy Volunteers | p[b] value |
| --- | --- | --- | --- | --- | --- | --- |
| | Total | Mask Positive | Mask Negative | p[a] value | | |
| Number | 29 | 13 (44.8) | 16 (55.2) | | 31.0 | |
| **Patient Characteristics** | | | | | | |
| Gender | | | | | | |
| Male | 26 (89.6) | 11 (84.6) | 15 (93.7) | 0.537 | 21 (67.7) | 0.059 |
| Female | 3 (10.3) | 2 (15.3) | 1 (6.25) | | 10 (32.2) | |
| Age, years Median (IQR) | 42 (32–52.5) | 44 (39–53) | 39 (30–51.75) | 0.232 | 42 (29–59) | 0.839 |
| 20–40 years | 11 (37.9) | 3 (23) | 8 (50) | 0.326 | 14 (45.1) | 0.34 |
| 41–60 years | 16 (55.1) | 9 (69.2) | 7 (43.7) | | 12 (38.7) | |
| >60 years | 2 (6.8) | 1 (7.6) | 1 (6.2) | | 5 (16.1) | |
| Comorbidities (Diabetes/Hypertension) | 10 (34.4) | 5 (38.4) | 5 (31.2) | 0.684 | 4 (12.9) | **0.048** |
| **COVID-19 Characteristics** | | | | | | |
| Antigen Positivity at Diagnosis | 15 (51.7) | 10 (76.9) | 5 (31.2) | **0.025** | NA | |
| Median (IQR) NPS Ct of N gene if rRT-PCR+ at Diagnosis | 30 (27.5–33.5) | 27 (26–28) | 32 (29.5–34) | 0.059 | NA | |
| Median (IQR) NPS Ct of N gene if rRT-PCR+ at Sampling | 29 (24–31) | 26 (21–29.5) | 30.5 (28–32) | **0.005** | NA | |
| **Contact History** | | | | | | |
| No Known Contact | 15 (51.7) | 5 (38.4) | 10 (62.5) | 0.273 | NA | |
| Known Contact (Family Member or Colleague) | 14 (48.2) | 8 (61.5) | 6 (37.5) | | NA | |
| **Symptoms** | | | | | | |
| Median (IQR) Number of Days since Onset of First Symptom | 5 (3–8) | 3.5 (3–7.5) | 5 (3–8) | 0.490 | NA | |
| Sore Throat | 13 (44.8) | 6 (46.1) | 7 (43.7) | 1.000 | NA | |
| Fever (all) | 23 (79.3) | 13 (100) | 12 (75) | | NA | |
| High Fever | 7 (24.1) | 6 (46.1) | 1 (6.2) | **0.016** | NA | |
| Mild Fever | 18 (62) | 7 (53.8) | 11 (68.7) | | NA | |
| No Fever | 4 (13.7) | 0.0 | 4 (25) | | NA | |
| Cough | 21 (72.4) | 10 (76.9) | 11 (68.7) | 0.696 | NA | |
| Breathing Difficulty | 14 (48.2) | 6 (46.1) | 8 (50) | 1.000 | NA | |
| Loss of Smell/Taste | 14 (48.2) | 7 (53.8) | 7 (43.7) | 0.715 | NA | |
| GI Symptoms (Loose Stools, Nausea) | 6 (20.6) | 3 (23) | 3 (18.7) | 1.000 | NA | |
| Weakness/Body ache/Headache | 10 (34.4) | 4 (30.7) | 6 (37.5) | 0.624 | NA | |
| Median (IQR) Number of Symptoms | 4 (3–5) | 4 (3–5) | 3 (2.2–5) | 0.384 | NA | |
| **COVID-19 Disease Status** | | | | | | |
| Mild | 18 (62) | 8 (61.5) | 10 (62.5) | 0.973 | NA | |
| Moderate without Pneumonia | 6 (20.6) | 2 (15.3) | 4 (25) | | NA | |
| Moderate with Pneumonia | 5 (17.2) | 3 (23) | 2 (12.5) | | NA | |
| **Drugs** | | | | | | |
| Doxycycline | 17 (58.6) | 7 (53.8) | 10 (62.5) | 0.289 | NA | |
| Ivermectin | 17 (58.6) | 8 (61.5) | 9 (56.2) | 1.000 | NA | |
| Azithromycin | 1 (3.4) | 1 (7.6) | 0 | 0.448 | NA | |
| Favipiravir | 10 (34.4) | 5 (38.4) | 5 (31.2) | 0.714 | NA | |
| Cephalosporin | 26 (89.6) | 11 (84.6) | 15 (93.7) | 0.573 | NA | |
| Hydroxychloroquine | 4 (13.7) | 2 (15.3) | 2 (12.5) | 1.000 | NA | |
| **Mask Sampling Characteristics** | | | | | | |
| Median (IQR) Sampling Score | 7 (5.5–8) | 8 (5.5–8) | 6 (5.2–7) | 0.131 | 7 (7–8) | **0.028** |
| **Sampling Preference** | | | | | | |
| Only Mask | 26 (89.6) | | | | 22 (70.9) | |
| Both Mask and Nasopharyngeal Swab | 0 | | | | 7 (22.5) | |

*(Continued)*

**Table 1.** (Continued)

| | NPS Positive (n = 29)* | | | | Healthy Volunteers | |
|---|---|---|---|---|---|---|
| Descriptions | Total | Mask Positive | Mask Negative | p$^a$ value | | p$^b$ value |
| Only Nasopharyngeal Swab | 2 (6.8) | | | | 1 (3.2) | |
| Neither Mask nor Nasopharyngeal Swab | 1 (3.4) | | | | 1 (3.2) | |

*Excludes 2 swab negative mask negative, Data are no. (%) of subjects, unless otherwise indicated.

Abbreviations: NPS- Nasopharyngeal Swab, IQR- Interquartile range, Ct- Cycle Threshold, rRT-PCR- Real time reverse transcriptase polymerase chain reaction.

p$^a$ Mask Positives Vs Mask Negatives; p$^b$ NPS positives (total) Vs Healthy Volunteers; p value significant at p<0.05-Significant p value highlighted in bold.

indicating that the intensity of the performance of tasks may not have affected the virus output in respiratory particles in this sampling. Moreover, we found no correlation between the human RnaseP Ct value (an indicator of sampling quality) and mask Ct value for E gene or sampling score (Supplementary Fig S3 in S1 File). The distribution of sampling score and associated mask Ct value for E gene in all patient samples is also shown in Supplementary Fig S4 in S1 File.

We next analyzed variations in the viral copies in mask positive patients based on the SARS-CoV-2 E gene (Supplementary Fig S2 in S1 File). Fig 1 displays the spatial distribution of SARS-CoV-2 virus viral load (A) and Ct values (B) in these patients, showing two distinct groups–(i) low emitters—mask positive patients with less than 100 viral copies expelled in 30 minutes (median 52.89, IQR 27.80–74.21) and (ii) high emitters- patients with > 1000 viral copies expelled in 30 minutes (median 2269, IQR 1421–16411) (Fig 1A). High emitters constituted only 30% (4/13) of the total mask positive patients and 12.9% of the total patients enrolled. Interestingly, such distinction was not observed when Ct values of NPS were considered. When the viral load was compared with days since onset of symptoms (Fig 1C), it was found that the low emitters had come in later in the infection stage for diagnosis- median 6 days (IQR 3–8 days) since symptom onset vs median 3 days (IQR 2.6–4.5) in high emitters, although the difference was not significant. Moreover, considering only the reported active infectious period of ≤5 days from onset of symptoms, [17] both high and low emitters were observed within this period and high emitters constituted 23% (4/17) of those patients (boxed data in Fig 1C), suggesting that stage of infection may not be the only contributing factor for low viral load. It may also be noted that there were a considerable number of mask negatives (9/17) within the 5 days' infection period. Other characteristics like sampling quality (sampling score; 8.5 for high emitters and 7 for low emitters; p = 0.08), age, contact, etc. also did not show variation between low and high emitters (Supplementary Table S1 in S1 File).

## Discussion

COVID-19 control strategies can be effectively implemented if there is a better understanding of how and by whom the virus is transmitted. However, little is known about the SARS-CoV-2 virus-laden particles generated by the patients during regular vocal and respiratory activities like talking, coughing, and breathing. Our study describes a potentially low-cost method using easily available materials to facilitate the detection and quantification of SARS-CoV-2 in respiratory particles expelled by patients during these activities in 30 minutes. This study shows that the expelled virus can be detected only in a subset of individuals (45%) who had confirmed diagnosis for COVID-19 by NPS based rRT-PCR. The results indicate that while mask-based sampling is not appropriate for use in the diagnosis of COVID-19, it may be a useful method to quantify transmission risks. The results are similar to those of a recent study by another group that investigated the SARS-CoV-2 virus in hospitalized severe COVID-19 patients in an

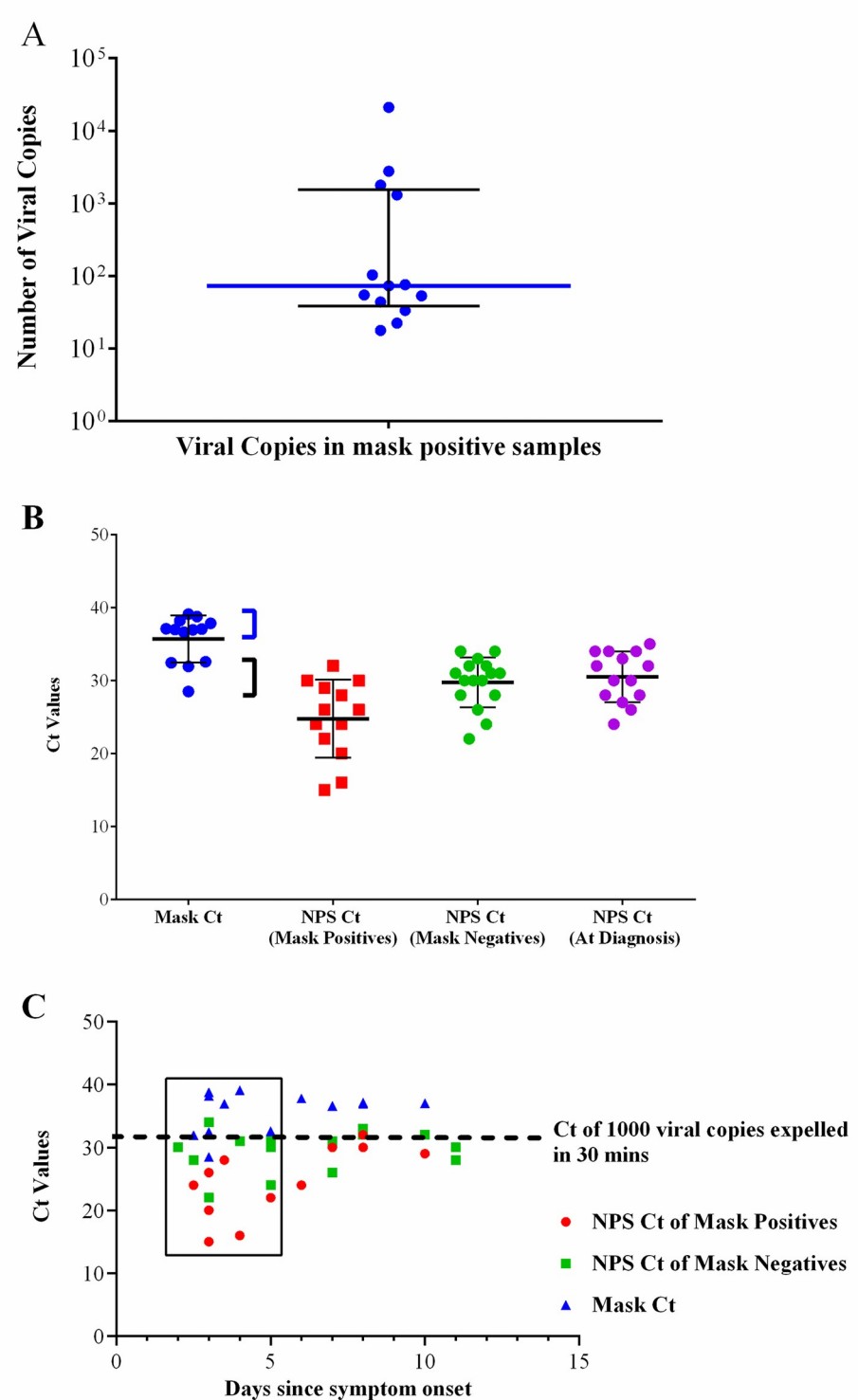

**Fig 1. Viral copies and Ct values in mask and NPS samples.** (A) SARS-CoV-2 viral copies expelled in 30 minutes by the mask positive patients. Data represented as median with IQR with the blue line indicating the median viral copies. (B) The distribution of Ct values from mask and NPS samples. The Ct value of the E gene in mask samples (blue) at sampling, the Ct value of the N gene in mask positive samples (red) and mask negative samples (green) at sampling, and Ct value of the N gene in patient samples at diagnosis. The mask E gene Ct values showed two distinct groups of samples with low Ct values (black bracket) and samples with high Ct values (blue bracket). No distinct groups were

seen in the N gene Ct values of NPS samples at enrollment or diagnosis. Data represented as median with IQR with the thick black line indicating the median Ct value. (C) Scatter plot with the Ct values of The E gene in mask and N gene in NPS patient samples on the Y-axis and days from onset of first symptoms of each patient on the X-axis. The mask E gene Ct values represented as blue triangles, the NPS N gene Ct values in mask positive patients, and mask negative patients represented as red dots and green squares respectively. The box encloses all the Ct value of the mask and NPS patient samples up to 5 days from the first onset of symptoms. The dotted line represents the Ct value when 1000 viral copies are expelled by the patients in 30 minutes. Abbreviations Ct- Cycle Threshold, NPS-Nasopharyngeal Swab.

older age group and observed an almost 40% positivity rate and an association between virus detection in respiratory particles with the severity of the disease [18]. The current study however could not explain this association to severity as all the enrolled patients were younger (median age 42) and with mild to moderate disease. Instead, this study describes the potential to measure the infectiousness of COVID-19 patients with mild/moderate disease through detection and quantification of viral load in respiratory particles expelled by patients and discusses its implications and relevance to transmission of the virus in the community.

In the absence of a reliable marker for transmission, viral load based on swab Ct is considered as a marker of infectiousness i.e. patients carrying high viral load/low Ct are likely to transmit more. This study shows that the NPS Ct values of mask positive patients were significantly lower than those of mask negative patients, indicating that patients with a higher viral load in their upper respiratory tract generally may emit more viruses and hence potentially be more infectious than mask negative patients. However, interestingly, not all low swab Ct (<30) yielded mask positivity and vice versa. The transmission of SARS-CoV-2 is known to be over- dispersed [3] like many other infectious diseases and a viral load based on swab Ct values may not satisfactorily explain this heterogeneity [19–21]. A recent epidemiological study describing the transmission of COVID-19 in two states of India with high prevalence observed that 70% of the patients yielded zero secondary infections among contacts [22]. Similar studies in China, Hong Kong, and Israel showed that most secondary infections (80%) arose from a small subset (8–20%) of the infected individuals [23–25]. Modelling studies have concluded that transmission is very unlikely (~0.00005%) when viral load is below $10^5$ RNA copies [26]. In congruence with these studies, the current study shows that the virus can be detected in respiratory particles of only 45% of the NPS positive patients and within these mask positive patients, there is a distinctly bimodal distribution of high and low emitters (Fig 1A). The high emitters constituted 12.9% overall and 23% of the patients captured in the known infectious period (within 5 days of symptom onset [17,21,27]). The bimodal distribution in emission patterns also have been identified in other airborne pathogens like influenza [9]. A similar distribution in the patient data was not observed in NPS Ct values (Fig 1B) suggesting that mask results and not NPS Ct depict variation among patients in terms of respiratory output, potentially reflecting the heterogeneous spread of COVID-19.

Another important supportive evidence for mask results reflecting the infectiousness of patients comes from studies that looked at the replication-competent virus from COVID-19 patients. Studies have shown that replication-competent live virus could not be detected in patients with Ct values above 24 to 34 in NPS samples and a large number of patients with lower Ct values (<24) also do not produce replication-competent virus [28–30]. Similarly in this study, we observed that the highest NPS Ct value beyond which mask positivity could not be observed was 32 for the N gene and conversely, several patients who had Ct values less than 30 in their NPS samples were also mask negative. In addition, we also observed that the viral load was not more than 100 copies in all mask positive patients who were diagnosed after 5 days of symptom onset. This is consistent with a published study that showed that the probability of finding infectious viruses decreases from about 40% at 5 days to <5% by 8 days after

symptom onset. Various epidemiological studies have also shown that secondary infections are almost nil among contacts if they had come in contact with the index case 5–7 days after symptom onset [17,31].

All of the above cumulatively suggest that the detection of SARS-CoV2 in respiratory particles using masks may prove to be useful in assessing the true infectiousness status of the COVID-19 patients and help in identifying high-risk contacts. Although it was interesting to note this relationship, the study has its limitations. The observations were based on small sample size and the study did not measure infections among contacts to establish infectivity or carry out longitudinal sampling within the same patients that may have helped in correlating it to true infectiousness. Moreover, the detection of the virus is still rRT-PCR based, which cannot differentiate replication-competent/infectious and non-replicating/non-infectious viruses.

Nevertheless, this study raises important questions that may be relevant for disease control efforts like intense contact tracing, reallocation of meagre resources, and prolonged containment. The availability of evidence of the type gathered in this study can provide opportunities to identify transmitters and hence may mitigate the need for one fits all infection control measure [32]. Mina and colleagues [32] suggest using antigen positivity results to focus on contact tracing efforts as a resource conservation measure. The results here show that respiratory particle positivity of the virus is significantly associated with antigen positivity (Table 1) and hence supports the idea that such an approach is likely to benefit the disease control efforts.

In conclusion, this study has shown the feasibility of detecting SARS-CoV-2 virus in respiratory particles expelled by patients using a simple collection method that may be used for assessing transmission risks of hosts, at different time points and during different activities. It would be interesting to study if a mass community screening using simple non-invasive mask sampling points to true transmission rates from symptomatic and asymptomatic individuals. It may also be insightful to probe the differences in the virus and host that contribute to heterogeneity in viral aerosol output and transmission. Pursuing these research questions may help us to understand the current pandemic as well as prepare ourselves for future pandemics.

## Supporting information

**S1 File.**
(DOCX)

## Acknowledgments

We thank Ms. Smriti Vaswani and Ms. Tejal Mestry (Research Assistants, FMR) for excellent technical contributions in processing patient mask samples; and Dr. Meet Visaria and Dr. Aishvarya Singh (Clinical Research Interns, Kasturba Hospital for Infectious Diseases) for diligently undertaking patient recruitment and sample collection at Kasturba hospital for infectious diseases; Mr. Nilesh Shahasne (Field Assistant, FMR) for sample transport between collaborating institutes. We would also like to acknowledge altona Diagnostics India Private Limited for the kind gift of SARS-CoV-2 Positive control IVT RNA for determining the viral load. Finally, we would like to thank all the participants of this study for their cooperation, patience, and support without which this study would not have been possible.

## Author Contributions

**Conceptualization:** Kalpana Sriraman, Ambreen Shaikh, Nerges Mistry.

**Data curation:** Kalpana Sriraman, Ambreen Shaikh.

**Formal analysis:** Kalpana Sriraman, Ambreen Shaikh.

**Funding acquisition:** Nerges Mistry.

**Investigation:** Kalpana Sriraman, Ambreen Shaikh, Nirjhar Chatterjee, Nerges Mistry.

**Methodology:** Kalpana Sriraman, Ambreen Shaikh, Jayanthi Shastri, Nerges Mistry.

**Project administration:** Kalpana Sriraman, Ambreen Shaikh, Swapneil Parikh, Shreevatsa Udupa.

**Resources:** Jayanthi Shastri.

**Supervision:** Jayanthi Shastri, Nerges Mistry.

**Writing – original draft:** Kalpana Sriraman, Ambreen Shaikh, Swapneil Parikh.

**Writing – review & editing:** Kalpana Sriraman, Ambreen Shaikh, Swapneil Parikh, Shreevatsa Udupa, Nirjhar Chatterjee, Nerges Mistry.

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
