## [Decision Letter · Decision Letter 0]

5 Feb 2021

PONE-D-20-37005

Non-invasive adapted N-95 mask sampling captures variation in viral particles expelled by COVID-19 patients: Implications in understanding SARS-CoV2 transmission

PLOS ONE

Dear Dr. Mistry,

Thank you for submitting your manuscript to PLOS ONE. After careful consideration, we feel that it has merit but does not fully meet PLOS ONE’s publication criteria as it currently stands. Therefore, we invite you to submit a revised version of the manuscript that addresses the points raised during the review process.

Please address all of the comments by the two reviewers, in particular by presenting more data on duration and type of vocal activities before resubmission. 

We look forward to receiving your revised manuscript.

Kind regards,

Joël Mossong

Academic Editor

PLOS ONE

Journal Requirements:

Reviewers' comments:

Reviewer's Responses to Questions

**Comments to the Author**

1. Is the manuscript technically sound, and do the data support the conclusions?

Reviewer #1: Yes

Reviewer #2: Partly

2. Has the statistical analysis been performed appropriately and rigorously? 

Reviewer #1: Yes

Reviewer #2: Yes

3. Have the authors made all data underlying the findings in their manuscript fully available?

Reviewer #1: Yes

Reviewer #2: Yes

4. Is the manuscript presented in an intelligible fashion and written in standard English?

Reviewer #1: Yes

Reviewer #2: Yes

5. Review Comments to the Author

Reviewer #1: Dear authors,

Many thanks for this interesting study. I would however, raise a few points

1) How was the sample size calculated?

2) Line 73-75

Please include the following reference:

https://pubmed.ncbi.nlm.nih.gov/32629023/

3) Line 110, 118: Please write the full form of "lab"

4) How was the duration for which the patient needed to wear the mask was standardised?

5) The authors could kindly give the details of the gelatine membrane used for the study.

Reviewer #2: In this manuscript, Sriraman et al. describes the use of a modified N-95 masks with a gelatin membrane, and recovered SARS-CoV-2 RNA in exhaled breath from about 40% of patients with mild/moderate COVID-19. They concluded that their results suggest there is variation in the emission of SARS-CoV-2 virus which may explain the heterogeneity in transmission risk between individuals.

The manuscript was clear overall. I have one major query: in lines 104-107 it described that various vocal tasks were performed during the 30 minutes of collection. Was only one vocal task performed for each sampling (apparently it was not, judging from the median sampling score of 6-8 in Table 1?), and how did the sample collector assign which vocal task to be performed (e.g. by randomisation)? This has significant impact on the interpretation and conclusions of the results shown, as the heterogeneity in viral load between individuals demonstrated may due to difference in the vocal tasks assigned. Please add a description of the number of participants assigned to each group in the Results section, and also provide a supplementary figure on the viral load stratified by different groups of vocal tasks (could be more than 3 groups as various intensity of each vocal tasks i.e. talking/coughing/breathing were assigned). The assignment of sampling score for each activity also seemed arbitrary, for example low talking, intermittent coughing and shallow breathing all shared the same weight of 1, although it would be expected low talking and intermittent coughing (in addition to breathing while in between talking/coughing) would shed more virus than shallow breathing.

Please find other minor suggestions, below:

- line 83: provide reference for the statement 'Ct value can indicate the potential infectiousness of different patients' (e.g. van Kampen et al Nat Commun 2021)

- lines 102-4: suggest to provide a supplementary figure to illustrate the mask sampling set-up

- line 162: please describe the sample type which the rapid antigen test at diagnosis was performed on

- line 201: Figure 1 legend - 'spatial' distribution is a misnomer?

- line 227: using the mask as an 'ideal method to quantity transmission risks' would underestimate the transmission risks via other routes of transmission?

- lines 233-235: may be could suggest that the present results would inform infectiousness of COVID-19 patients with mild/moderate illness, which would have a more relevant interpretation on the transmission risk in the community (compared to severe cases who would be hospitalised)?

- lines 255-257: although there is heterogeneity in exhaled breath viral shedding, should also express some uncertainty on whether it is directly related to heterogeneity in transmission as transmission can be via other routes

- references: please confirm and update preprints that is published (e.g. ref # 1 is published in JAMA Network Open)

6. PLOS authors have the option to publish the peer review history of their article (what does this mean?). If published, this will include your full peer review and any attached files.

Reviewer #1: No

Reviewer #2: No

---

## [Author Response · Author response to Decision Letter 0]

23 Feb 2021

AUTHORS’ REPSONSE TO REVIEWERS’ COMMENTS

We thank the reviewers for their positive and encouraging comments on the study. We have responded to the queries and trust that they are clarified substantially.

Reviewer#1: 

Comment- Many thanks for this interesting study. I would however, raise a few points

How was the sample size calculated?

Response - The primary design objective of the study was to check the ability of mask sampling to detect SARS-CoV-2 RNA and evaluate concordance with the standard nasopharyngeal swab method. Since the test outcome is binary and we were interested in calculating the proportion of samples positive for mask against 100% positive standard samples, we applied the proportion test for sample size calculation. We estimated that a minimum of thirty samples were required to check if the proportion of mask positive samples matches with the standard test (https://www.benchmarksixsigma.com/calculators/sample-size-calculator-for-1-proportion-test/). The assumptions for the calculation were 95% significance, 80% power and 10% acceptable difference. We took an equal number of healthy volunteers to verify concordance in known negative standard samples.

The manuscript methods section has been now revised to include the sample size calculation method in Lines 98-100 and now reads as 

“The sample size was calculated using a proportion test for binary outcome with assumptions of 95% confidence interval, 80% power and 10% acceptable difference.”

Comment- Line 73-75, Please include the following reference:

https://pubmed.ncbi.nlm.nih.gov/32629023/e

Response - Thank you for the suggestion. We have now included the reference as No 13 in the aforementioned place.

Comment - Line 110, 118: Please write the full form of "lab"

Response - Thank you for pointing out. We have now written the full form in the revised manuscript.

Comment - How was the duration for which the patient needed to wear the mask was standardized?

Response - This method was primarily standardized in TB patients for detection of TB bacteria where 10 minutes was selected based on the yield and stability of TB RNA (Shaikh et al 2019, Reference No 14 in the revised manuscript). Considering minimum sampling time for patients’ convenience, we tested the same conditions as earlier (10 minutes) and 30 minutes in a small pre-pilot of 4 patients each. The 30 minutes was chosen based on other patient air sampling studies that tested viruses using a 20-30 minute protocol (Ref Nos 9, 10, 12 in the revised manuscript). Our initial results showed better concordance at 30 minutes (4/4 as against 1/4) and hence 30 minutes was fixed as mask sampling time.

Comment - The authors could kindly give the details of the gelatine membrane used for the study.

Response - As mentioned in the Methods section we used commercially available 37mm gelatin membrane filter from Sartorius, Gottingen, Germany. The catalog number of the product is 12602-37-ALK. The Fig S2 depicts the N95 mask lined with gelatin membrane.

Reviewer #2: 

Comment - In this manuscript, Sriraman et al. describes the use of a modified N-95 masks with a gelatin membrane, and recovered SARS-CoV-2 RNA in exhaled breath from about 40% of patients with mild/moderate COVID-19. They concluded that their results suggest there is variation in the emission of SARS-CoV-2 virus which may explain the heterogeneity in transmission risk between individuals. The manuscript was clear overall. 

Response - Thank you for the positive comment

Comment - I have one major query: in lines 104-107 it described that various vocal tasks were performed during the 30 minutes of collection. Was only one vocal task performed for each sampling (apparently it was not, judging from the median sampling score of 6-8 in Table 1?), and how did the sample collector assign which vocal task to be performed (e.g. by randomisation)? This has significant impact on the interpretation and conclusions of the results shown, as the heterogeneity in viral load between individuals demonstrated may due to difference in the vocal tasks assigned. 

Response – Participants did not perform only one task. Each participant performed all vocal tasks as directed by the sample collector in a particular order viz: The participants were asked to carry on with the activities whatever they were doing for the first 20 minutes and then undertook certain purposeful vocal tasks in the last 10 minutes as directed by the collector. The sequence of the purposeful tasks was as follows

1. Talk or Read - 3 mins

2. Cough 20 times- (1 minute)

3. Deep breath for 1 minute

4. Talk or Read-3 mins

5. Cough 20 times- (1 minute)

6. Deep breath for 1 minute

Since a standard procedure involving all aforementioned tasks were followed and no differences existed in tasks assigned for any participant, there was no randomization necessary to group the individuals based on task. 

The collector instructed the patients to talk or read aloud, cough forcefully and perform deep breathing. Although specific instructions were given, the intensity of the task varied between patients. Hence the collector subjectively noted the actual intensity with which the participant performed each task and recorded it in the case record form which was used to measure the quality of sampling. The case record questionnaire had the following format which was used by the sample collector to note the intensities of the tasks performed.

1. Participant compliance information and experience with mask sampling: (Please tick the appropriate option)

a. While sampling, 

i. Task1 : Talked/Read/Sang/recited prayer/Recited poem

1. Volume of Task 1: Loud/Normal/Low

ii. Task 2 Coughing : Intermittent/Continuous

1. Task 2 Coughing Intensity: Light/Deep and forceful

iii. Task 3 Breathing: 

1. Shallow/ Deep

b. Post Sampling, participant felt easier and comfortable with

Mask sampling/Swab Sampling

We agree that if the participant had performed different tasks or either of the task, the variation would have impacted the output viral load. In this study, we used a standardized task approach to minimize the variation that could affect the sampling and viral output. All patients performed the same tasks for the same length of time. Moreover, we did not observe any correlation between the human RNase P Ct levels in the samples (considered generally as an indicator of sample quality) and mask Cts for E gene (R2= 0.1603) or sampling score (R2=0.003) suggesting that the viral output was independent of the amount of total RNA collected from the patients (please see graphs below). Lastly, as mentioned in the results section (Lines 187-190 in the unmarked revised manuscript), there was no association between mask results or Ct with sampling score.

We have now explained the sampling process in detail in the revised manuscript methods section (Lines 107-116, 120-123 of the unmarked revised manuscript) and now reads as

107-116 - “The participants were asked to carry on with the activities whatever they were doing for the first 20 minutes and undertook certain purposeful vocal tasks in the last 10 minutes. The purposeful tasks included following tasks in sequence as directed by the sample collector.

i. Talk or Read - 3 mins

ii. Cough 20 times- (1 minute)

iii. Deep breath for 1 minute

iv. Talk or Read-3 mins

v. Cough 20 times- (1 minute)

vi. Deep breath for 1 minute

120-123- During mask sampling, the sample collector subjectively noted the actual intensity with which, each participant performed the vocal task and recorded the details in the questionnaire format of the case record form (Supplementary information-mask sampling section).

We have also added a line on the estimation of RnaseP in the materials and method section along with the complete description in supplementary (Lines 127-130 of the unmarked revised manuscript) and now read as 

127-130- A retrospective analysis of human RnaseP gene, an indicator of sample quality was carried out in all mask samples using Taqpath SARS-CoV-2 detection kit V1 (Details in supplementary information) 

Also, we have added the appropriate lines describing RnaseP sampling results and its absence of correlation with mask Ct values of E gene and sampling score in results section, with the analysis and graphs shown above, added to the supplementary section (Fig S2). The lines 187-192 of the unmarked revised manuscript now read as - 

187-192 - Moreover, we found no correlation between the human RnaseP Ct value (an indicator of sampling quality) and mask Ct value for E gene or sampling score (Supplementary Fig S3). The distribution of sampling score and associated mask Ct value for E gene in all patient samples is also shown in supplementary Fig S4.

Comment - Please add a description of the number of participants assigned to each group in the Results section, and also provide a supplementary figure on the viral load stratified by different groups of vocal tasks (could be more than 3 groups as various intensity of each vocal tasks i.e. talking/coughing/p were assigned). 

Response - As mentioned in the above point, since each participant performed all tasks, the participants cannot be stratified based on the tasks performed. To illustrate the point of sampling score and mask results, we have now added a supplementary Fig S4 with a graph depicting the Ct E gene and sampling score.

Comment - The assignment of sampling score for each activity also seemed arbitrary, for example low talking, intermittent coughing and shallow breathing all shared the same weight of 1, although it would be expected low talking and intermittent coughing (in addition to breathing while in between talking/coughing) would shed more virus than shallow breathing. 

Response - We agree that we have not made direct output measurements and the sampling score was assigned based on the intensity of each task with assumptions made from literature. Studies have shown that the number of particles emitted increases with the loudness of voice and varies with velocities and the number of times the tasks are performed (Asadi et. al. 2019, Bake et al 2019, Wilson et. al 2020). There are several studies available that looked at size distribution and output with various tasks (Fennelly,2020). Based on the literature, we assigned increasing numbers to the increasing intensity of the task. We used the following table to calculate the sampling score with the lowest suggesting low particle output and the highest score suggesting maximum particle output. Based on aerosol dynamic knowledge available, cumulatively we expected intensity of combined tasks would relate to particle output and hence recoverable virus particles. We agree that this is only a suggestive and not a precise estimate. More comprehensive studies would be required to tease out what type of task and conditions would contribute to viral particle emission by infected individuals and how it relates to transmission.

Task Intensity Assigned score 

Talking/reading Loud voice 3

 Normal voice 2

 Low voice 1

Coughing Deep and forceful and continuous coughing 

4

 Deep and forceful but intermittent coughing 3

 Light and continuous coughing 2

 Light, and intermittent coughing 1

Breathing Deep breathing 2

 Shallow breathing 1

Asadi, S., Wexler, A.S., Cappa, C.D. et al. Aerosol emission and super emission during human speech increase with voice loudness. Sci Rep 9, 2348 (2019). https://doi.org/10.1038/s41598-019-38808-z

Bake, B., Larsson, P., Ljungkvist, G. et al. Exhaled particles and small airways. Respir Res 20, 8 (2019). https://doi.org/10.1186/s12931-019-0970-9

Wilson, N.M., Norton, A., Young, F.P. and Collins, D.W. (2020), Airborne transmission of severe acute respiratory syndrome coronavirus‐2 to healthcare workers: a narrative review. Anesthesia, 75: 1086-1095. https://doi.org/10.1111/anae.15093

Comment - Please find other minor suggestions, below: - line 83: provide reference for the statement 'Ct value can indicate the potential infectiousness of different patients' (e.g. van Kampen et al Nat Commun 2021)

Response - The reference has been now included as suggested.

Comment - lines 102-4: suggest to provide a supplementary figure to illustrate the mask sampling set-up

Response - A picture of the mask with membrane has been provided in supplementary document (Fig S2) as suggested.

Comment - line 162: please describe the sample type which the rapid antigen test at diagnosis was performed on

Response - The test was performed using nasopharyngeal swab as recommended by manufacturers. The change has been made in the revised document in Line 176 and now reads as 

“Mask positivity in patients was associated with higher rapid antigen test positivity in NPS samples at diagnosis (p=0.025)”

Comment - line 201: Fig 1 legend - 'spatial' distribution is a misnomer?

Response - The words spatial distribution has been removed and the legend now reads as

“SARS-CoV-2 viral copies expelled in 30 minutes by the mask positive patients.”

Comment - line 227: using the mask as an 'ideal method to quantity transmission risks' would underestimate the transmission risks via other routes of transmission?

Response - The word ideal has been now been replaced by word useful and the lines 248-249 in revised manuscript now reads as 

“The results indicate that while mask-based sampling is not appropriate for use in the diagnosis of COVID-19, it may be a useful method to quantify transmission risks.”

Comment - lines 233-235: maybe could suggest that the present results would inform infectiousness of COVID-19 patients with mild/moderate illness, which would have a more relevant interpretation on the transmission risk in the community (compared to severe cases who would be hospitalized)?

Response - We agree that the information generated from mild and moderate cases would be more relevant to community transmission and have been discussed in detail in the manuscript. Based on the suggestion, we have revised the lines in the manuscript now to bring more stress to that aspect and lead the reader to a detailed discussion in the Discussion section. The lines now read as 

Lines 253-258- “The current study however could not explain this association to severity as all the enrolled patients were younger (median age 42) and with mild to moderate disease. Instead, this study describes the potential to measure the infectiousness of COVID-19 patients with mild / moderate disease through detection and quantification of viral load in respiratory particles expelled by patients and discusses its implications and relevance to transmission of the virus in the community.”

Comment - lines 255-257: although there is heterogeneity in exhaled breath viral shedding, should also express some uncertainty on whether it is directly related to heterogeneity in transmission as transmission can be via other routes

Response - So far the evidence for other routes of transmission has been shown as rare, though not completely negated. We have now revised the sentence to reflect this uncertainty. The lines now read as 

Lines 278-280 “A similar distribution in the patient data was not observed in NPS Ct values (Fig 1B) suggesting that mask results and not NPS Ct depict variation among patients in terms of respiratory output, potentially reflecting the heterogeneous spread of COVID-19”

Comment - references: please confirm and update preprints that is published (e.g. ref # 1 is published in JAMA Network Open)

Response - Thank you for pointing it out. We have now revised the reference and also checked all preprints for their publication status and revised it accordingly. Following are the references that were revised based on the current status of publications.

1. Madewell ZJ, Yang Y, Longini IM, Jr, Halloran ME, Dean NE (2020) Household Transmission of SARS-CoV-2: A Systematic Review and Meta-analysis. JAMA Network Open 3: e2031756-e2031756.

2. van Kampen JJ, van de Vijver DA, Fraaij PL, Haagmans BL, Lamers MM, et al. (2021) Duration and key determinants of infectious virus shedding in hospitalized patients with coronavirus disease-2019 (COVID-19). Nature communications 12: 1-6.

---

## [Decision Letter · Decision Letter 1]

22 Mar 2021

Non-invasive adapted N-95 mask sampling captures variation in viral particles expelled by COVID-19 patients: Implications in understanding SARS-CoV2 transmission

PONE-D-20-37005R1

Dear Dr. Mistry,

We’re pleased to inform you that your manuscript has been judged scientifically suitable for publication and will be formally accepted for publication once it meets all outstanding technical requirements.

Kind regards,

Joël Mossong

Academic Editor

PLOS ONE

Additional Editor Comments (optional):

Reviewers' comments:

Reviewer's Responses to Questions

**Comments to the Author**

1. If the authors have adequately addressed your comments raised in a previous round of review and you feel that this manuscript is now acceptable for publication, you may indicate that here to bypass the “Comments to the Author” section, enter your conflict of interest statement in the “Confidential to Editor” section, and submit your "Accept" recommendation.

Reviewer #1: All comments have been addressed

Reviewer #2: All comments have been addressed

2. Is the manuscript technically sound, and do the data support the conclusions?

Reviewer #1: Yes

Reviewer #2: (No Response)

3. Has the statistical analysis been performed appropriately and rigorously? 

Reviewer #1: Yes

Reviewer #2: (No Response)

4. Have the authors made all data underlying the findings in their manuscript fully available?

Reviewer #1: Yes

Reviewer #2: (No Response)

5. Is the manuscript presented in an intelligible fashion and written in standard English?

Reviewer #1: Yes

Reviewer #2: (No Response)

6. Review Comments to the Author

Reviewer #1: (No Response)

Reviewer #2: Thank you for addressing my comments and the additional details on the sampling procedure. Regarding the sampling score, I would suggest to add a brief sentence in the Discussion section commenting the arbitrary nature of the assignment of the sampling score, and be more cautious when making this conclusion of "The variation in the sampling score was not significant, indicating that the intensity of the performance of tasks may not have affected the virus output in respiratory particles in this sampling".

7. PLOS authors have the option to publish the peer review history of their article (what does this mean?). If published, this will include your full peer review and any attached files.

Reviewer #1: No

Reviewer #2: No

---

## [Editor Report · Acceptance letter]

24 Mar 2021

PONE-D-20-37005R1 

Non-invasive adapted N-95 mask sampling captures variation in viral particles expelled by COVID-19 patients: Implications in understanding SARS-CoV2 transmission 

Dear Dr. Mistry:

I'm pleased to inform you that your manuscript has been deemed suitable for publication in PLOS ONE. Congratulations! Your manuscript is now with our production department. 

Kind regards, 

on behalf of

Dr. Joël Mossong 

Academic Editor

PLOS ONE